# Culturable Yeast Diversity of Grape Berries from *Vitis vinifera* ssp. *sylvestris* (Gmelin) Hegi

**DOI:** 10.3390/jof8040410

**Published:** 2022-04-16

**Authors:** Gustavo Cordero-Bueso, Ileana Vigentini, Roberto Foschino, David Maghradze, Marina Ruiz-Muñoz, Francisco Benitez-Trujillo, Jesús M. Cantoral

**Affiliations:** 1Department of Biomedicine, Biotechnology and Public Health, University of Cádiz, 11009 Cádiz, Spain; marina.ruiz@uca.es (M.R.-M.); jesusmanuel.cantoral@uca.es (J.M.C.); 2Department of Food, Environmental and Nutritional Sciences, University of Milan, 20122 Milan, Italy; ileana.vigentini@unimi.it (I.V.); roberto.foschino@unimi.it (R.F.); 3Department of Agriculture, Faculty of Viticulture and Winemaking, Caucasus International University, 0141 Tbilisi, Georgia; david.maghradze@gmail.com; 4Department of Mathematics, University of Cádiz, 11510 Puerto Real, Spain; quico.benitez@uca.es

**Keywords:** *Saccharomyces cerevisiae*, species dominance, species richness, wild vine, yeast biodiversity

## Abstract

*Vitis vinifera* L. ssp. *sylvestris* (Gmelin) Hegi is recognized as the dioecious parental generation of today’s cultivars. Climatic change and the arrival of pathogens and pests in Europe led it to be included on the International Union for Conservation of Nature (IUCN) Red List of Threatened Species in 1997. The present work focused on the study of culturable yeast occurrence and diversity of grape berries collected from wild vines. Sampling was performed in 29 locations of Azerbaijan, Georgia, Italy, Romania, and Spain. In total, 3431 yeast colonies were isolated and identified as belonging to 49 species, including *Saccharomyces cerevisiae*, by 26S rDNA D1/D2 domains and ITS region sequencing. Isolates of *S. cerevisiae* were also analyzed by SSR–PCR obtaining 185 different genotypes. Classical ecology indices were used to obtain the richness (S), the biodiversity (H’), and the dominance (D) of the species studied. This study highlights the biodiversity potential of natural environments that still represent a fascinating source of solutions to common problems in winemaking.

## 1. Introduction

Populations of the Eurasian wild grapevine *Vitis vinifera* L. subspecies *sylvestris* (Gmelin) Hegi are spread across the Caucasus region, Mediterranean Basin, Iberian Peninsula, and as far as the Hindu Kush and the Maghreb. Their cultivars belong to *Vitis vinifera* L. subspecies *vinifera*, and they are the dioecious parental generation of today’s cultivars [1]. Wild grapes are predominantly climbers in natural forests and river basins and occur in disjunct populations. They grow in very special habitats where microclimatic conditions are present [2]. They occasionally form complex introgressive hybrid groups in transition zones nearby vineyards with cultivated grapes [3]. While wild grapevine is dioecious with anemophilous pollination, domesticated grapevine is self-pollinating (hermaphrodite); moreover, both differ in terms of several traits (e.g., sugar content, berry size, and form) [4,5]. Palynological evidence of pollen from the genus *Vitis* in the Middle Pleistocene was found in the ‘El Padul’ peat bogs in Granada, Spain [6], and in the river basin at ‘Laguna de las Madres’ in Huelva, Spain [7]. Wild grapevine fossils (half a million years old) were discovered in Azerbaijan in excavations near Nakhchyvan [8]. According to Rivera and Walker [9] and Maghradze et al. [10], wild grape berries have been a human food source in the Caucasus area and on the Italian and Iberian Peninsulas since the Neolithic age. Thus, the domestication process of wild vine was initially driven and controlled by cultural concerns; only afterward were the customs of colonizers such as the Phoenicians and Greeks introduced into the daily routine of people living in the Western Mediterranean [5]. It can be assumed that the introduction of winemaking and later the viticulture practices of the invaders intersected with those of existing local cultivation, already at the advanced stage of selecting the most suitable *V. vinifera* ssp. *sylvestris vines*. Forni [11] and Arroyo-García et al. [12] evaluated various chlorotypes and their distribution in thousands of samples of diverse wild and cultivated vines, coming from different areas of the Iberian and Italian Peninsulas, the Caucasus region, and the north of Africa. Their results reinforced the theory of the polycentric origin of the domestication of the vine, and more than 70% of the cultivated vines through the Iberian Peninsula displayed chlorotypes compatible with wild grapevine communities placed in Western Europe [12]. Unfortunately, urbanization and industrialization of territories, climatic change, and the arrival in Europe of pathogens and pests led *Vitis vinifera* ssp. *sylvestris* (Gmelin) Hegi to be included on the IUCN Red List of Threatened Species in 1997. Even though numerous studies about the biogeographical distribution [2,13,14,15], ampelographic characterization [16,17], tolerance to pests and diseases [18], and the current health status of this species have been carried out [12,16], to the best of our knowledge, no studies on the microbial populations of grape-berries have been conducted.

It is well known that yeasts are part of the natural communities of grapes [19,20]. Thus, grape berries are always considered a potential source of new wine yeasts. A countless number of studies covering microorganisms associated with grape berries from *Vitis vinifera* ssp. *vinifera* cultivars have shown that the occurrence and diversity of yeast populations are determined by physiological, anthropogenic, and environmental factors [21,22], emphasizing the *terroir* concept of microbial components. Actually, the mycobiota is naturally shaped by vineyard location [23,24,25,26], vintage and climatic conditions [27,28], grape variety [22,29], and the ripeness and health status of the grape berries [21,30]. Human activities also have a high impact on yeast diversity and distribution, such as farming system type [31,32], pest management [33,34], soil irrigation, and vineyard maintenance [35]. On the other hand, few studies have addressed the yeast communities present in grapes and fermenting musts deriving from *Vitis* non-*vinifera* ecosystems [27,36]. In particular, some studies [37,38] recently provided preliminary evidence on the specific association between *Vitis labrusca* species and some yeast species and strains, indicating that specific biological interactions might underlie the vine plant–microorganism associations.

We previously reported in a congress proceeding the preliminary results of this study, as well as its possible scientific impact [39]. However, to the best of our knowledge, no new studies focusing on the culturable yeast occurrence and diversity in the carposphere of *V. vinifera* ssp. *sylvestris* from the Caucasus area and Mediterranean Basin (29 different locations of Azerbaijan, Georgia, Italy, Romania, and Spain) have been published. Final outputs of this study will allow us to (i) obtain primary and precise information about yeast communities on berries of wild vines, (ii) provide an objective framework for the classification of the broadest range of species according to their extinction risk, (iii) select attractive yeast strains for their biotechnological potential, offering new opportunities to winemakers, and (iv) set up a collection of wild strains with enological origin in order to define novel domestication genetic targets for future evolutionary studies in wine yeasts.

## 2. Materials and Methods

### 2.1. Sampling Plan, Fermentation, and Yeast Isolation Procedures

Field investigations were carefully planned, choosing habitats with appropriate ecological characteristics for wild grapevine: well separated from cultured vineyards, proximity to rivers, soil with a high degree of humidity, and a fairly good conservation level of vegetation [2]. Grape bunches were collected from 29 different wild populations of the species *Vitis vinifera* ssp. *sylvestris* in Azerbaijan (two sampling sites), Georgia (nine), Italy (10), Romania (one), and Spain (seven) during the 2013 to 2016 vintages. An ampelographic description of the Spanish samples was carried out by an expert group from “Instituto Madrileño de Investigación y Desarrollo Rural, Agrario y Alimentario” (Madrid, Spain). Italian and Romanian samples were determined at the Department of Agricultural and Environmental Sciences of the University of Milan (Milan, Italy). Azerbaijani and Georgian samples were described in the Department of Viticulture of the Institute of Viticulture and Winemaking (Tbilisi, Georgia). Grapevines were analyzed and characterized according to the criteria proposed by the OIV (1997).

The maturity stage of the grapes during sampling was determined first by following the protocol proposed by Coombe [40] and then in the field by simply applying a drop of three grape berries (randomly selected) onto the prism of a manual portable refractometer Master Series (ATAGO, Tokio, Japan). Subsequently, the mean of each measurement was calculated. Measurements were expressed as °Brix. Afterward, the samples characterized by a °Brix lower than 20 were considered not ripe, while the samples with a °Brix higher than 20 were considered to be ripe.

Approximately 0.5–1.0 kg of healthy grape bunches, depending on the year and the sampling location, were collected from the different vine populations (Appendix A) under aseptic conditions, placed into sterile bags, transported to the laboratory in portable refrigerators, and processed within 3 h. The isolation of yeasts was achieved using two different procedures: by directly removing the epiphytic and endophytic microbial cells from berry skins just after the harvest or after a natural enrichment of grape juice through a spontaneous fermentation with maceration.

For the former case, 100 berries were selected from each grapevine using disinfected scissors and distributed in several 50 mL Falcon tubes before adding 0.9% (*w*/*v*) sodium chloride solution and 0.2% (*v*/*v*) Tween-80 to dislodge the epiphytic microorganisms. Samples were incubated at 30 °C for 2 h with agitation (150 rpm) and were then treated by sonication (Vibra-Cell 75185, Sonics and Materials Inc., Newtown, CT, USA) for 3 s at medium power amplitude (55%).

To isolate culturable endophytic yeasts from the grape tissues, at least another 50 grape berries with good sanitary status were selected from each sampled bunch. A sterilization method with some modifications [41] was used to suppress epiphytic fungi. Grapes were first washed with deionized water 5–6 times, then immersed in 70% ethanol for 1 min and in 2.5% sodium hypochlorite for 10 min, and finally washed again 5–6 times with sterile distilled water. Ground tissues were then removed with a sterile scalpel following the protocol of Isaeva et al. [42] with some modifications. Fragments of internal storage tissues were removed and suspended in 15 mL tubes containing 5 mL of a solution of sodium chloride (0.9% NaCl) and 0.2% (*v*/*v*) Tween-80. Tubes were agitated with a vortex mixer at maximum speed for 5 min.

In the latter case, for the detection of minority species that would not be detected by direct plating, an enrichment step repressing the propagation of alcohol-sensitive yeasts was also included in the process. Grape bunches, stems excluded, were crushed and homogenized by means of a Stomacher^®^ Biomaster 80 (Seward GmbH, Worthing, UK) to obtain an adequate volume of grape juice. Then, 250 mL of sample was poured into 500 mL sterile fermenters equipped with two openings, one at the top for filling and the other in the middle for sampling operations. The two holes were sealed with rubber stoppers, and the top one was equipped with a capillary tube to allow the CO_2_ to flow off. Spontaneous fermentation was carried out at 20 °C without the addition of sulfur dioxide. The pH and °Brix values of the musts were monitored by a pH meter (Crison GLP21, Barcelona, Spain) and refractometer (Atago digital refractometer model CO., LTD. Tokyo, Japan), respectively. Samples were taken at the intermediate point, when their weights were reduced by 70 g/L (sugar consumption), and at the end of fermentation to evaluate the yeast communities, as stated by [43]. 

Lastly, decimal dilutions of the solutions of the sonicated grape berries and the vortexed grape tissues, as well as the fresh musts and one-third of the fermented wine (self-enrichment) samples, were spread onto Wallerstein Laboratory Nutrient Agar (WL, Condalab, Madrid, Spain) to evaluate colony color and morphology [44]. Plates were incubated at 28 °C, examined daily for colony growth, and counted. Thirty colonies from each plate were randomly selected, when possible, considering the colony morphology. Selected colonies were then spread onto YGC Agar culture medium (0.5% yeast extract, 2% *w*/*v* glucose, 0.01% chloramphenicol, 1.5% agar–agar) to avoid bacterial contaminations. Both epiphytic and endophytic isolates were considered to be isolated from grape berry skins. 

After appropriate isolation, pure cultures were stored at −80 °C in YPD broth (1% *w*/*v* yeast extract, 2% *w*/*v* peptone, 2% *w*/*v* glucose) with added glycerol (25% *v*/*v*), or for short-term storage at 4 °C on YPD medium plus agar (2% *w*/*v*).

### 2.2. Species Identification 

DNA extraction from yeast colonies was carried out using the protocol suggested by [45], and DNA samples were stored at −20 °C. A UV/Vis spectrophotometer (Nanodrop 1000, Thermo Fisher Scientific Inc., Waltham, MA, USA) was used to calculate the quantity of DNA extracted. Yeast identification was performed by PCR amplification of the internal transcribed spacers between the 18S and 26S rDNA genes (ITS1–5.8S–ITS2) using primers ITS1 and ITS4 [46] according to the related protocol. Subsequent restriction analysis (RFLP) of the amplified products was conducted according to [47] using *Cfo*I, *Dde*I, *Hae*II, *Hinf*I, and *Taq*I restriction enzymes (Thermo Fisher Scientific Inc.). Amplified products and their restriction fragments were separated on 1.4% (*w*/*v*) and 2.5% (*w*/*v*) agarose gels, respectively, and stained with a final concentration of 0.05 µL/mL ethidium bromide, in 1× TBE buffer at 100 V for 90 min. DNA fragment sizes were determined by comparison with a molecular ladder marker of 100 bp (Promega, Madison, WI, USA). 

Strains identified as *Saccharomyces cerevisiae* were also subjected to Multiplex-Mi https://wi.knaw.nl/ (accessed on 1 March 2022) protocol of [48] for MyTaq™ DNA Polymerase (Bioline, Toronto, ON, Canada). At least two isolates from each ITS-RFLP genotype group were randomly selected for sequencing ITS1–5.8S–ITS2 and the LSU rRNA gene D1/D2 domain. ITS1–5.8S–ITS2 was PCR-amplified as mentioned above. Sequences of certain species such as *Aureobasidium pullulans* and *Rhodotorula nothogafi* have identical D1/D1 sequences to other species. Thus, when necessary, we included the ITS1–5.8S–ITS2 region sequences. Amplification of the D1/D2 region was carried out using primers NL1 and NL4, as previously described [46]. Purification and sequencing of PCR products were performed by Macrogen Inc. facilities (Seoul, South Korea) using an ABI3730 XL automatic DNA Analyzer. The obtained sequences were aligned using the ClustalX algorithm. The Basic Local Alignment Search Tool (BLAST) (https://www.ebi.ac.uk/Tools/sss/ncbiblast/, accessed on 1 March 2022) was used to compare the sequences obtained with databases from the European Molecular Biology Laboratory (EMBL). As stated by Sipiczki [30], the GenBank entries are not checked for the correctness of their taxonomic assignment by the depositors. The D1/D2 sequences of the isolates were then aligned with the D1/D2 sequences of the type strains of the species whose sequences were found to be highly similar in the GenBank search. For this, the sequences of the type strains were downloaded from the CBS database (https://wi.knaw.nl/, accessed on 1 March 2022). The sequence similarity with the type strain sequences was also determined by pairwise BLAST alignment using the bl2seq algorithm available on the NCBI website. We considered identification as correct when the sequence showed an identity ≥98% and a good query cover.

### 2.3. Statistical Analysis

Classical ecology indices were used to evaluate the overall biodiversity analysis. The species richness (S), Shannon–Wiener index (H′), and Simpson index (D) were applied as stated by Cordero-Bueso et al. (2011). Due to the different number of samples collected in each region and the different species richness in the analyzed regions, Hill’s number, specifically the number of abundant species (N1), the number of very abundant species (N2), and evenness (E’) were also calculated [49]. Both data processing and analysis were carried using R software version 3.6.3 [50]. Coordinate data were also processed to generate a simple geographic map using R.

Regarding the microsatellite analysis of those isolates belonging to *S. cerevisiae*, the genetic diversity indices, i.e., the mean number of alleles (Na), number of effective alleles (Nae), observed heterozygosity (Ho), expected heterozygosity (He), Shannon’s information index (I), Nei’s genetic identity and distance (D), and among population pairs, were estimated by GenALEx 6.5 [51].

## 3. Results

### 3.1. Sampling Plan and Overall Biodiversity Analysis

We sampled 29 different grapevine populations throughout territories belonging to five different countries of the Eurasian area on a W–E cline spanning approximately 6000 km, from Tui (Spain) to Quba (Azerbaijan) (Figure 1). Coordinates are shown in Appendix A. The mean distance between *V. vinifera* ssp. *sylvestris* populations within Italy and Spain was 800 km, while it was 500 km within the Georgia and Azerbaijan local regions sampled. Only one location was sampled in Romania. It is important to note that eight wild vine populations were sampled in Sardinia (Italy), which is an island with a distinct microclimate. All samples were collected between September and December, depending on the latitude and the ripening status of the grape berries, between 2013 and 2016. Not all years enabled the collection of grapes from the same vine plant due to the dioecious character of *V. vinifera* ssp. *sylvestris*, drought, bird attacks, fungal infections of grapes, or logistic concerns. Because of this, it was decided to group all isolates into six regions: Azerbaijan, Georgia, northern Italy, Romania, Sardinia (Italy), and Spain (Figure 1).

In total, 3431 culturable yeast colonies were isolated from the collected grape bunches at each sampling point (Appendix A). Taking into account the colonies isolated from grape berries and fresh musts, a total of 1660 isolates were obtained (1563 as epiphytic and 97 as endophytic yeast isolates). Regarding the isolates from self-enrichment, 1771 yeast colonies were isolated, as shown in Appendix A.

Results of molecular identification using ITS1–5.8S–ITS2 amplification and restriction analysis showed 49 different patterns. The choice of appropriate restriction endonucleases is critical for RFLP experiments. The commonly used *Cfo*I, *Hae*III, and *Hinf*I enzymes failed to segregate *M. guilliermondii* from other species of the same genus. Indeed, *Meyerozyma guilliermondii* (anamorph *Candida guilliermondii*) and *Meyerozyma caribbica* (anamorph *Candida fermentati*) are closely related species. Thus, to avoid misidentification, these yeasts were subjected to RFLP analysis using the enzyme *Taq*I, as stated by [52]. The D1/D2 region of the 26S rDNA of the two yeast strains for each species presumptively identified by RFLP was also sequenced to confirm such identification. *Aureobasidium pullullans* can easily be confused with other similar species such as *Aureobasidium subglaciale*, *Kabatiella microsticta*, or *Columnospaeria fagi* because many database D1/D2 sequences of these species are identical [30]. Moreover, *R. nothofagi* is difficult to distinguish from *C. pallidicorallinum* because certain database D1/D2 sequences of these species are identical [30]. Since the matching patterns of the type strains of these species exhibited the most similar ITS sequences and the most similar D1/D2 sequences, it was justified to assign the yeast strains of this study to *A. pullulans* and *R. nothofagi* [30].

On the other hand, we faced problems whereby some of the isolates that seemed to belong to *Metschnikowia*-like strains did not show D1/D2 sequence identity with any of the type strains despite being fairly similar to one species of the *Metschnikowia pulcherrima* clade. This also happened with the ITS sequences. In agreement with [53,54,55], species belonging to the so-called *M. pulcherrima* clade cannot be unequivocally assigned to one of the species after rDNA analysis because some species such as *M. fructicola* or *Metschnikowia andauensis* have a non-homogenized rDNA array. When we compared the sequences of our isolates identified as *Metschnikowia* sp. with the sequences deposited in the Mycobank database, the most probable species related to this genus were *Metschnikowia chrysoperlae* (similarity of 99.43%) and *Metschnikowia pulcherrima* (similarity of 99.20%). Thus, we decided to maintain these sequences as *Metschnikowia pulcherrima*-like species. This was also the case for the *Hanseniaspora* sp. species, which matched with *H. uvarum* (lower than 89% similarity) in the Mycobank database. GenBank accession numbers for all species identified in each country are provided in Appendix A.

Once the culturable isolates were identified, overall biodiversity analysis was carried out, grouping them in six regions (Table 1). Regarding the yeast richness (S), 25 different species were isolated in Spain, 20 were isolated in Sardinia, 15 were isolated in northern Italy, and 11 were isolated in Georgia, while four and two species were found in Romania and Azerbaijan, respectively. Taking into account Shannon’s index (H’) and the concentration of dominance (D), Spain was the most diverse country since it had the lowest concentration of dominance and the highest diversity (D = 0.11, H’ = 2.52). Nevertheless, there was a case of discordance in Italy since the concentration of dominance was slightly lower in northern Italy, while Shannon’s index was higher in Sardinia (D = 0.19, H’ = 2.04; D = 0.20, H’ = 2.15, respectively). In order to correctly evaluate this discordance, Hill’s numbers were calculated to determine the effective number of species. Thus, Sardinia was found to be more diverse than northern Italy (N1 = 8.58, 7.72; N2 = 5.01, 5.29, respectively).

On the other hand, the lowest diversity was found in Azerbaijan, since only two different species were detected, and the concentration of dominance was very high (D = 0.74, H’ = 0.43). It is interesting to note that Romania was found to be much more diverse than Azerbaijan (D = 0.30, H = 1.28), although only one sample was collected in Romania. In addition, the proportion of the different species found in the six regions, evaluated by Hill’s evenness, showed that Romania and Azerbaijan were the most equitable (E’ = 0.92, E’ = 0.88, respectively), showing the codominance of the few species found in these two regions, while the relatively high evenness together with the high number of effective species in Spain highlights the equitability of the species found in these samples (E’ = 0.72, N1 = 12.49, N2 = 8.97), followed by Italy (E’ = 0.69, N1 = 7.72, N2 = 5.29 in northern Italy; E’ = 0.58, N1 = 8.58, N2 = 5.01 in Sardinia).

### 3.2. Occurrence and Distribution of Wild Culturable Yeasts Isolated from Grapes and Fresh Musts

All of the species isolated from berry skins and fresh musts taken from each location were grouped by country (except northern Italy and Sardinia), taking into account both the relative abundance of each species (Figure 2A) and their presence in each region (Figure 2B). Thus, a total of 39 different species belonging to 22 genera were identified. Among them, only three species were found with a relative high frequency; *Hanseniaspora uvarum* was the most abundant (12.4%), followed by *Pichia kluyveri* (11.5%) and *Pichia kudriavzevii* (10.6%). The rest were found with a lower frequency, presenting some unique species in the different regions analyzed except for Romania, where no unique species was found. Thus, *Saccharomycodes ludwigii* was isolated only in Azerbaijan, while the species *Clavispora lusitaniae* and *Rhodotorula mucilaginosa* appeared only in Georgia. On the other hand, northern Italy was the only region where the species *Candida californica*, *Curvibasidium cygneicollum* (anamorph *R. fujisanensis*), *Pichia occidentalis*, and *Cryptococcus flavescens* were isolated, while, in Sardinia, the species *Aureobasidium proteae* and *Metschnikowia viticola* were unique. Moreover, in Sardinia, yeasts of the *S. cerevisiae* species were not isolated in musts and grape skins (Figure 2B). It should be noted that the species *Curvibasidium pallidicorallinum* and *Rhodotorula nothofagi* were only isolated in the two regions of Italy. Lastly, in Spain, which is also where most samples were obtained, a total of 15 unique species were found (Figure 2B). It is interesting to note that 97 colonies were isolated as endophytic microorganisms inside the grape berry skin tissues. Among those, 11 different yeast species were identified (Appendix A). The predominant endophytic species were *S. cerevisiae* and *H. guilliermondii*, followed by *Zygosaccharomyces fermentati* and *M. pulcherrima (*Appendix A).

### 3.3. Occurrence and Distribution of Wild Yeasts Isolated from Self-Enrichment Musts

The musts of the selected grapes from each region were allowed to spontaneously ferment to obtain a natural self-enriched medium in yeast populations until the sugar content was less than 3 g/L. All spontaneous fermentations were correctly achieved. A total of 1771 yeast colonies were isolated at both the intermediate point and the end of the process, as evaluated by the consumption of sugars and the release of carbon dioxide (Figure 3). From these isolates, 721 belonged to yeasts of non-*Saccharomyces* genera and 1050 belonged to the species *S. cerevisiae* (Appendix A). Thus, a total of 33 species belonging to 20 genera were identified. In this case, the relative abundance of the isolated species was quite different (Figure 3A); *Saccharomyces cerevisiae* represented almost half of the total isolates (48%), along with a considerable frequency of the species *H. uvarum* (11.2%) and *P. kluyveri* (8.6%). The other species were found in a frequency lower than 5%.

Regarding the species found in each region (Figure 3B), in this case, no unique species were found in Azerbaijan or in Romania. In Georgia, the species *C. lusitaniae* and *I. terricola* were unique, while, in northern Italy, only *Martiniozyma asiatica* was exclusive. In both Sardinia and Spain, a greater number of exclusive species were found. In Sardinia, the presence of species *Hanseniaspora clermontiae* (which was not found in musts), *Filobasidium wieringae*, *C. pallidicorallinum*, *R. nothofagi*, *Pichia manshurica*, and *Zygosaccharomyces* sp. should be highlighted. On the other hand, in Spain, six unique species were found: *C. californica*, *C. sake*, *Lachancea thermotolerans*, *M. caribbica* (and the anamorph *M. guillermondii*), *Starmerella bacillaris*, and *Starmerella stellata*. Furthermore, it should be noted that *Hanseniaspora* sp. was also found in both Spain and Sardinia after self-enrichment (Figure 3B).

### 3.4. Intraspecific Analysis of Isolates Belonging to Saccharomyces cerevisiae

Since many isolates belonged to the species *S. cerevisiae*, a total of 1127 isolates were genotyped by microsatellite multiplex PCR analysis amplifying the SC8132X, YOR267C, and SCPTSY7 loci [48,56]. After genotyping, a total of 154 isolates were sequenced for these three loci amplified. An inter- and intra-population analysis was then carried out by grouping the different genotypes found in the six regions described above.

The genetic diversity of populations, obtained through molecular markers, was measured mainly by values related to allelic richness and heterozygosity indices (Table 2), as all samples were polymorphic. For the dataset obtained, a relative allelic diversity was detected at the three loci studied on the basis of the following genetic diversity indices: mean number of alleles (Na), number effective alleles (Nae), and Shannon’s information index (I). Georgia was the region with the highest number of alleles and with the greatest diversity (Na = 25.7, Nae = 14.3, I = 2.9 on average at all three loci), followed by Spain (Na = 21, Nae = 11.4, I = 2.7 on average at all three loci). On the other hand, Azerbaijan and Romania showed the lowest values, possibly due to the low number of samples collected (Na = 5.3, 4.3; Nae = 4.3, 4.1; I = 1.5, 1.4, respectively, on average). The heterozygosity indices were highly variable, with mean values ranging from 0.5 (Azerbaijan and Sardinia) to 0.9 (northern Italy). However, the most striking result was that none of the pairs of observed and expected heterozygosity indices coincided, with the observed value being lower than the expected one in almost all cases (except for northern Italy). Therefore, there was a deficiency of heterozygous individuals in the populations analyzed.

For the inter-population analysis, Nei’s genetic distance and pairwise Fst were used in order to evaluate the genetic structure. Although the genetic distances between populations were low, Georgia, Sardinia, and Spain shared a greater genetic diversity (D = 0.95, 0.84, and 0.73, respectively). In addition, this group of higher genetic diversity was also described using the Fst statistic (0.031, 0.032, and 0.023, respectively). Fst also showed that northern Italy was related to this group, albeit with a slightly smaller difference (0.039 with Georgia, 0.043 with Sardinia, and 0.035 with Spain). However, AMOVA showed that there were no significant differences between the genetic populations of *S. cerevisiae* from the six studied regions.

## 4. Discussion

Wild grapevines (*V. vinifera* ssp. *sylvestris*) throughout the Eurasian region are a relatively unexploited source of yeasts. In order to increase knowledge about their biodiversity, grape berries were collected from these vines from a total of 29 sampling points located in five countries in four different years (2013–2016). Due to the very nature of the study, as mentioned above, it was not possible to collect samples from all sampling points every year. Therefore, it was decided to group the results into six regions (Azerbaijan, Georgia, Northern Italy, Romania, Sardinia, and Spain) since it was found that there was no loss of information when calculating indices independently in each location of each region. It was not possible to correlate yeast diversity with the location of the wild vines (altitude and latitude) or with the different chlorotypes of wild vines found in the Eurasian region, related in turn to the origin of these vines [12].

There is a growing trend to study the whole microbial diversity through NGS, such as metabarcoding approaches. However, this work focused on the study of culturable yeast diversity since it can be used from both a biotechnological and an enological point of view. Hamad et al. [57] showed that ITS1/ITS2 amplicon sequencing provides different information about fungal communities compared to culturomics; nevertheless, both approaches are complementary, assessing fungal diversity. Accordingly, combining both culturomics and amplicon-based metagenomic approaches may be a promising strategy toward analyzing fungal compositions in an ecosystem.

Yeast identification was conducted applying the ITS-PCR and the RFLP techniques combined with ITS and LSU D1/D2 sequencing. This method helped to distinguish some misidentifications in some species, as stated in Section 3. Moreover, to solve these biases, we compared the D1/D2 and ITS sequences of the isolated strains with those of the strain types of the species in both NCBI and CBS databases. Moreover, we used the RFLP method to strengthen the correct identification, as the database curators occasionally did not match the taxonomic affiliation. It is important to point out that the strains of the species of the *pulcherrima* clade could not be well distinguished from each other using the criteria applied in this work. According to Sipiczki [54] the lack of reliable phenotypic differences and barcode gaps hampers the taxonomic identification of new isolates. Thus, taking into account the results in the present manuscript, the fact that the species of the *pulcherrima* clade should be merged into one species is in agreement with Sipiczki’s proposal [54].

This work considered that some of the species were found uniquely in some of the sampled regions. For instance, in Spain, 15 unique and highly represented species were found among all isolates. Thus, this isolation seems to be an unexpected event. In fact, because of the complexity mentioned above of the sampling procedure, not all regions were equally explored in depth by analyzing the identical number of samples/grapes/fresh musts in which the same unique species could be identified. Consequently, Hill’s numbers were also calculated to determine the effective number of species.

Non-*Saccharomyces* yeasts are known to be present in a variety of environmental niches. Wine yeasts are mainly isolated from grape surfaces and their musts [58]. Other places are also potential habitats for non-*Saccharomyces* wine-related species, such as the vineyard soil, sour rot-damaged grapes, the winery, and its equipment [58,59]. The vast majority of research papers focusing on yeast diversity considered the grape cultivars (conventional, organic, and biodynamic) as the main source of non-*Saccharomyces* yeasts [60]. On the other hand, a few studies isolated nonconventional yeasts from non-*vinifera* cultivars such as *Vitis labrusca*, *Vitis rotundifolia*, *Vitis amurensis*, and *Vitis davidii* [37,38,60,61]. To the best of our knowledge, this is the first study considering yeast isolation from *Vitis vinifera* ssp. *sylvestris* (Gmelin) Hegi.

A large occurrence and diversity of non-*Saccharomyces* yeast species can be isolated even before the self-enrichment process starts, e.g., during ripening and harvest processes. However, from the enological and biotechnological perspectives, some yeast species are usually present during the fermentation process. After processing, three groups of non-*Saccharomyces* yeasts can be found in the first stages of grape must fermentation [62], during which they proliferate due to their lower tolerance to ethanol: aerobic yeasts (e.g., yeasts of genera *Pichia, Rhodotorula*, and *Cryptococcus*), yeasts with low or medium fermentation ability (e.g., genera *Hanseniaspora* and *Metschnikowia*), and yeasts with fermentative capacity (*Lachancea, Zygosaccharomyces*, and *Torulaspora*). These yeasts are then gradually replaced by the ethanol-tolerant yeast *S. cerevisiae* [62,63].

Regarding the different species found, the group of apiculate yeasts of the genus *Hanseniaspora* represents an important proportion of the microbiota of wild grape berries, a result in accordance with isolations from other cultivars of *V. vinifera* [60]. *Hanseniaspora uvarum* was the predominant non-*Saccharomyces* species isolated from grapes, musts, and self-enrichments, but it was not the dominant species in all sampled areas. Other different species belonging to the genus *Hanseniaspora* were found, specifically *H. guillermondii* and *H. opuntiae* (both only in Spain) in musts, as well as *H. guillermondii* and *H. opuntiae* (found in both Spain and Sardinia), and *H. clermontiae* (only found in Sardinia), probably due to its latitude. On the other hand, patterns for the genera *Clavispora*, *Saccharomycodes*, and *Yarrowia* and the species *M. fructicola* and *P. fermentans* showed preference for grapes from the central–oriental Mediterranean Basin longitudes (Georgia, Azerbaijan, and Romania) [60]. In contrast, some genera had a higher number of occurrences in grapes from warmer areas such as Italy, Sardinia, and Spain: *Metschnikowia* sp., *L. thermotolerans*, *P. kluyveri*, *P. manshurica*, *Scheffersomyces stipitis,* and *S. bacillaris*. Although no clear and direct association could be established between any particular genus and a geographical location, some patterns could be confirmed as being associated with the climates where the grapes were collected. In light of the above information, the most representative non-*Sacharomyces* yeast genera occurring on wild grapes were comparable with those identified in hundreds of other reports dealing with grape microbial communities around the world. However, we found occasional occurrences of several species that were not previously reported in any other grape/wine-associated niche (e.g., *Schwanniomyces polymorphus*) [60,64,65].

Some of the isolated species, such as endophytic yeasts, were identified as *H. guilliermondii*, yeasts of the *M. pulcherrima* clade, *R. fujisanensis*, *R. mucilaginosa*, and yeasts of the genus *Cryptococcus*, which were previously described as isolated species from internal tissues of fleshy fruits in the study of Isaeva et al. [42]. Interestingly, the isolated yeast species not yet described in the literature such as *P. occidentalis*, *T. delbrueckii*, and yeasts of the genus *Zygosaccharomyces* were found in the internal tissue of some collected grape berries.

Furthermore, *S. cerevisiae* was the predominant species when self-enrichment was carried out, and it was isolated in all regions, as expected. Actually, *S. cerevisiae* was isolated in musts and grape skins from all regions except Sardinia, demonstrating that this species is found in grape samples. This finding contradicts other studies postulating that this species was found in low proportions or was even absent in grapes [30,66]. However, in this work, it was found in relatively considerable proportions in both grapes and musts, which agrees with other previous studies [25,31]. The source in this case comes from a species not domesticated by humans (*V. vinifera* ssp. *sylvestris*), in addition to being a wild vine; thus, these yeasts are necessarily found in nature, having developed different mechanisms for their survival. Accordingly, they should be deeply analyzed in terms of both potential and genetic diversity. The fermentative *S. cerevisiae* was also isolated as an endophytic yeast in all sampled areas except Sardinia. In this sense, Mandl et al. [67] proved the hypothesis that vines can take up yeasts from the soil and transport them through the vine to the stems and skins of grapes during the preharvest season. The yeast took longer than one week to move from the soil to the grapevine stem. After 11 days, during the second sampling, the commercial *S. cerevisiae* yeasts applied in the vineyard were detected. It is assumed that the speedy transport of yeasts through the xylem is similar to that of bacteria [68]. Despite collecting grapes from Sardinia at the correct maturity stage, these yeasts were probably picked up earlier. However, this is a hypothesis needing further investigation.

Microsatellite analysis carried out on the *S. cerevisiae* isolates showed that the three loci analyzed were polymorphic; however, this analysis was insufficient to deeply assess genetic diversity among populations. It is necessary to increase the number of loci to correctly analyze the allelic diversity within this species. In addition, the genetic diversity according to AMOVA was not significant among the six regions analyzed. However, the relative diversity found among samples can provide substantial information. It is important to consider that, despite the statistical analysis, there were no significant differences among the sampled areas, and *S. cerevisiae* populations associated with *V. vinifera* ssp. *sylvestris* seemingly showed a pattern of geographical differentiation, at large and small scales [69,70]. The heterozygosity values observed (Ho) were much lower than the expected heterozygosity (He). This phenomenon, constituting a heterozygosity deficit, is related to genetic drift and nonrandom mating between individuals of the populations, related in turn to migration [71]. However, it is today understood that *S. cerevisiae* mostly reproduces through budding or intra-ascus mating (both reducing the heterozygosity level of the population) in most natural environments. On the other hand, the population heterozygosity (not the individual one) appeared to be higher than that previously observed, providing an exciting perspective. Hence, the future plan is to increase the number of loci to analyze the allelic diversity within the *S. cerevisiae* species isolated in different regions, allowing a deeper comparison of the data with previous studies [25,72,73]. On the other hand, it was confirmed that the regions of Spain, Sardinia (Italy), and Georgia were those that shared the greatest genetic diversity. A correlation between genetic diversity and the regions analyzed, located more than 100 km from each other, could not be found. This was also demonstrated in a similar deep analysis of *S. cerevisiae* populations [25,74]. On the other hand, the population heterozygosity (not the individual one) was seemingly higher than that observed previously; hence, further studies should be performed to confirm this. It should be noted that other factors can modify the genetic variation of these yeasts in a certain niche. One such factor is represented by birds that can transport and diffuse yeasts and filamentous fungi on a large scale during migration across the Mediterranean Sea [75,76]. Microbial agents from migratory bird cloacae play a key role since they represent the last tract reached by microorganisms after gastric transit [76]. Conversely, insects can transport *S. cerevisiae* cells on a small scale, mainly wasps, bees, and flies [65,69,77]. These insects are attracted to certain volatile compounds that *S. cerevisiae* can produce, easily mobilizing them due to this mutualistic relationship [78] and, thus, promoting a migration that would expectedly break the genetic equilibrium.

Regarding the overall biodiversity analysis, Spain and Italy (both Sardinia and northern Italy) were the most diverse countries. It should be noted that these two countries are where the greatest number of samples were collected. However, to compensate for the difference in the number of isolates, the indices used to finally define diversity were Hill’s number and evenness, obtaining the same conclusion. The greater biodiversity found in these countries could be explained by the concept of migration. Spain and Italy have the largest areas of cultivated vineyards; accordingly, the migration flow of yeasts due to insects, mainly bees, wasps, and flies [79], may be greater than in the case of the other three countries analyzed, thus explaining the higher diversity found. This is related to what has been observed for the contribution of migratory birds in the dissemination of a large number of different yeast species during migration [75].

Some species found in the present work could also be of interest in biotechnological fields other than enology. Actually, certain isolates from the collection generated in this work, specifically belonging to the species *M. guillermondii*, *H. uvarum*, *H. clermontiae*, and *P. kluyveri*, have been described as antagonists of phytopathogenic fungi such as *Botrytis cinerea*; hence, they could be used to design biocontrol strategies [80]. In addition, there are other potential capabilities that could be worth exploiting, e.g., the production of biosurfactants with antifungal activity recently described in *R. paludigena* [81], the production of carotenoids by *R. glutinis* from different sources such as glycerol [82], the production of bioethanol from lignocellulosic residues due to the ability to ferment xylose [83], or the ability to act as growth promoters of different plants [84].

The results obtained in this work reveal the yeast biodiversity from a so far unexplored source and provide a collection of yeasts obtained from grapes from reservoirs of *Vitis vinifera* ssp. *sylvestris* distributed across different locations of the Eurasian region. Some of these yeast isolates, due to the very nature of their source, could have interesting biotechnological potential both in sustainable agriculture and in mitigating the effects of climate change on winemaking.

## Figures and Tables

**Figure 1 jof-08-00410-f001:**
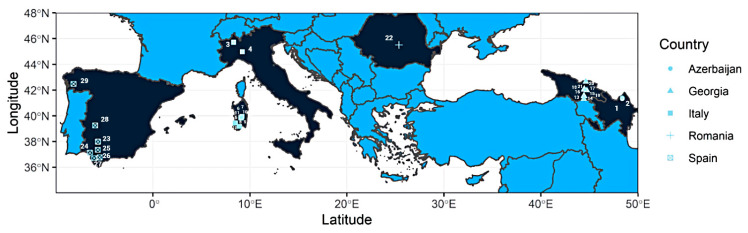
Locations of the sampled wild grape berries from *V. vinifera* ssp. *sylvestris* (Gmelin) Hegi. Sites from Azerbaijan; 1—Guruchay1, 2—Guruchay2, sites from Italy; 3—Monte Fenera, 4—Montalto, 5—Ortuabis, 6—Bau Sa Mela, 7—Santa Sofia, 8—Ristalu, 9—Fluminimaggiore Nera, 10—Fluminimaggiore Bianca, 11—Gutturu Mannu1, 12—Gutturu Mannu2, sites from Georgia; 13—Tsminda, 14—Nakhiduri23, 15—Tsitsamuri, 16—Zhinvali 03, 17—Shirikhevi 06, 18—Nakhiduri 24, 19—Shirikhevi 02, 20—Bagichala03, 21—Barisakmos, sites from Romania; 22—Turcul River, sites from Spain; 23—Hueznar, 24—La Rocina, 25—El Bosque, 26—Gran Capitán, 27—La Algaida, 28—Guadalupe, 29—Tui.

**Figure 2 jof-08-00410-f002:**
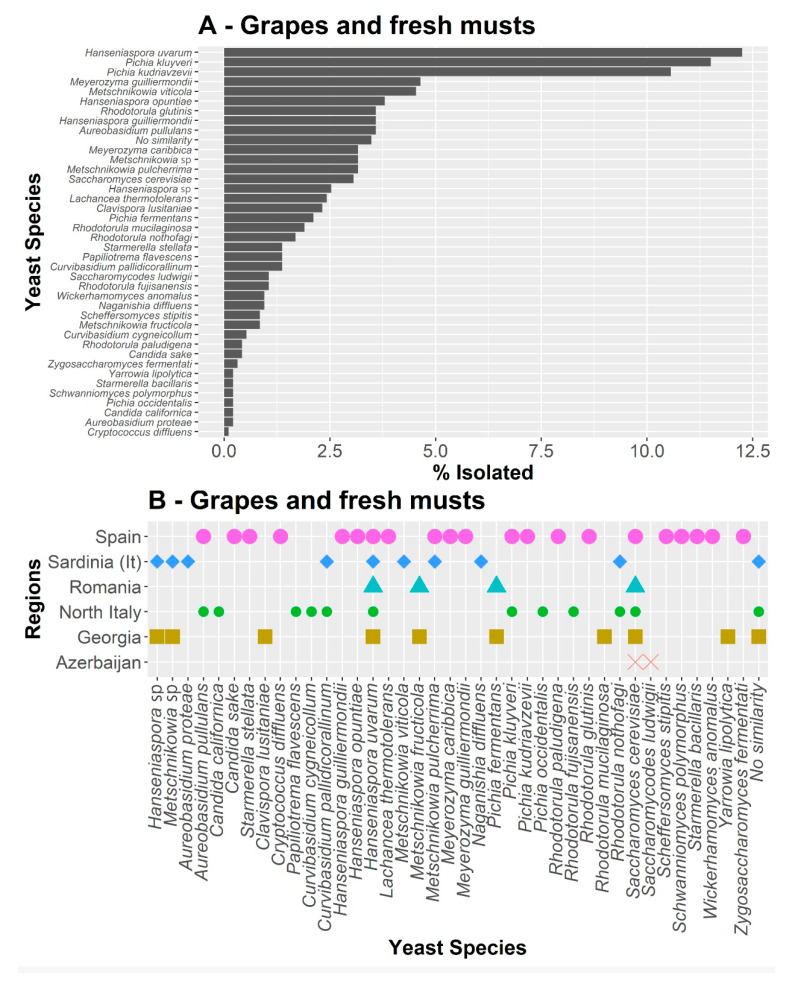
All culturable species isolated from berry skins and fresh musts taken from each location, taking into account both the relative abundance of each species (**A**) and their presence in each region (**B**).

**Figure 3 jof-08-00410-f003:**
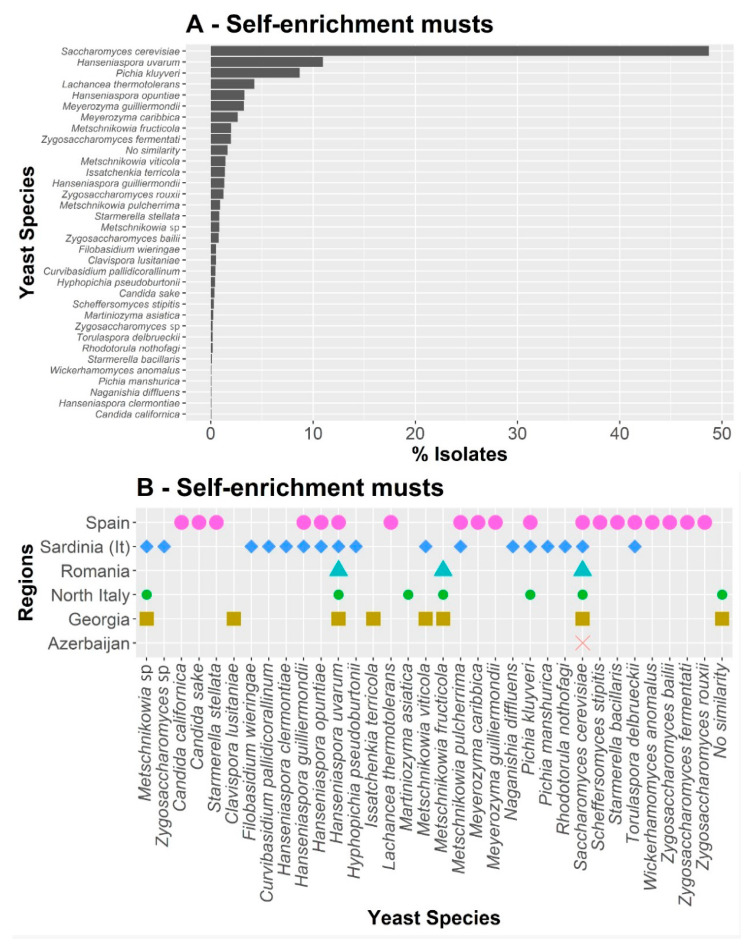
Distribution of culturable species isolated at both the intermediate point and the end of the spontaneous fermentation, as evaluated by the consumption of sugars and the release of carbon dioxide (**A**), and the species found in each region (**B**). In this case, no unique species were found in Azerbaijan or in Romania.

**Table 1 jof-08-00410-t001:** Ecological index analysis grouping the six regions (S = species richness; D = Simpson’s index; H = Shannon’s index; N1 = Hill’s abundant species index; N2 = Hill’s very abundant species index; E’ = Hill’s evenness index).

REGION	S	D	H	N1	N2	E’
Azerbaijan	2	0.74	0.43	1.54	1.35	0.88
Georgia	12	0.33	1.63	5.09	3.01	0.59
Northern Italy	15	0.19	2.04	7.72	5.29	0.69
Romania	4	0.30	1.28	3.59	3.32	0.92
Sardinia	20	0.20	2.15	8.58	5.01	0.58
Spain	25	0.11	2.52	12.49	8.97	0.72

**Table 2 jof-08-00410-t002:** Genetic diversity of populations measured by values related to allelic richness (N) and heterozygosity indices: number of alleles (Na), number of effective alleles (Nae), observed heterozygosity (Ho), expected heterozygosity (He), and Shannon’s information index (I).

REGION	Locus	N	Na	Nae	I	Ho	He
Azerbaijan	YOR267C-1	6	5	3.79	1.47	0.5	0.74
	SCPTSY7-1	5	7	5.56	1.83	0.6	0.82
	SC8132X-1	4	4	3.56	1.32	0.25	0.72
Georgia	YOR267C-1	72	26	17.17	3	0.63	0.94
	SCPTSY7-1	58	26	15.79	2.97	0.52	0.94
	SC8132X-1	70	25	9.88	2.67	0.54	0.9
Northern Italy	YOR267C-1	8	11	9.14	2.31	0.75	0.89
	SCPTSY7-1	8	11	9.14	2.31	0.88	0.89
	SC8132X-1	7	14	14	2.64	1	0.93
Romania	YOR267C-1	3	4	3.6	1.33	0.33	0.72
	SCPTSY7-1	3	6	6	1.79	1	0.83
	SC8132X-1	3	3	2.57	1.01	0.33	0.61
Sardinia	YOR267C-1	14	10	6.64	2.09	0.36	0.85
	SCPTSY7-1	13	13	8.05	2.31	0.54	0.88
	SC8132X-1	15	19	15.52	2.84	0.67	0.94
Spain	YOR267C-1	48	22	8.44	2.55	0.5	0.88
	SCPTSY7-1	49	21	11.63	2.69	0.69	0.91
	SC8132X-1	42	20	14.11	2.8	0.64	0.93

## Data Availability

Not applicable.

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
