# Peer review of "Culturable Yeast Diversity of Grape Berries from Vitis vinifera ssp. sylvestris (Gmelin) Hegi"

_jof, 2022, doi:10.3390/jof8040410_

Round 1

Reviewer 1 Report

In this study, the authors isolated more than 3400 yeast strains from grapes of a wild grapevine distributed in 29 locations in five Eurasian countries. A total of 49 species were identified from the yeast strains based on ITS RFLP combined with ITS and LSU D1/D2 sequence analyses. Furthermore, the genetic diversity of the Saccharomyces cerevisiae strains isolated was analyzed using SSR-PCR analysis, resulting in the identification of 185 different genotypes. Though the yeasts isolated are of potential value in winemaking and biotechnology, the novelty of this study is unclear and the strategy and methodology used in this study are problematic.

1. The sampling strategy is quite biased. The numbers of sites sampled in different regions differ greatly. The great difference in samples collected from different regions makes the comparison between different regions meaningless.

2. It is not clear how many vine plants were sampled at each site. The method description implies that only one sample (0.5-1.0 kg grape bunches) from one plant was sampled in each site. If it was true, the yeast diversity in each site was not fully revealed.

3. The yeast strains isolated were differentiated mainly based on ITS RFLP analysis. This is an outdated method as mentioned in this manuscript that some species cannot be distinguished based on the RFLP patterns. Since DNA sequencing is commonplace and cheap now, why the authors did not use ITS and D1/D2 sequencing as the main method for yeast strain differentiation and species identification?

4. Climate parameters were not measured and recorded; thus, it is unable to analyze the data in terms of ecology. It is not appropriate to call the data shown in Table 2 ecological indexes.

5. It is questionable to include the yeast strains isolated after enrichment incubation in biodiversity index calculation and comparison as shown in Table 2, because enrichment must alter the yeast diversity originally harbored in the samples.

6. Isolation of culturable endophytic yeasts from the grape tissues is described in Materials and Methods, but the result is not mentioned in the Results section.

Author Response

In this study, the authors isolated more than 3400 yeast strains from grapes of a wild grapevine distributed in 29 locations in five Eurasian countries. A total of 49 species were identified from the yeast strains based on ITS RFLP combined with ITS and LSU D1/D2 sequence analyses. Furthermore, the genetic diversity of the Saccharomyces cerevisiae strains isolated was analyzed using SSR-PCR analysis, resulting in the identification of 185 different genotypes. Though the yeasts isolated are of potential value in winemaking and biotechnology, the novelty of this study is unclear and the strategy and methodology used in this study are problematic.

Thanks to the reviewer for the comments. To better point out the novelty of this manuscript we have re-written the paragraph at the end of the introduction as follows; “Under the best of our knowledge no studies focused on the study of yeast occurrence and diversity on carposphere of V. vinifera ssp. sylvestris from the Mediterranean Basin (29 different locations of Azerbaijan, Georgia, Italy, Romania and Spain) have been published before. Final outputs will allow: i) to obtain primary and precise information about yeast communities on berries of wild vines; ii) to provide an objective framework for the classification of the broadest range of species according to their extinction risk; iii) to select attractive yeast strains for their biotechnological potential, offering new opportunities to winemakers; iv) to set up a collection of wild strains with oenological origin in order to define novel domestication genetic targets for future evolutionary studies in wine yeasts.”

On the other hand, we believe that strategy and methodology is well stablished elsewhere and in most of previous publications from the authors of the present work and others. Nevertheless, we will extend and clarify the methodology point by point in order to improve the M&M section.

  1. The sampling strategy is quite biased. The numbers of sites sampled in different regions differ greatly. The great difference in samples collected from different regions makes the comparison between different regions meaningless.

We agree with the reviewer’s comments, but samples were taken in natural environments where biotic and abiotic factors can’t be avoided. Wild vines are usually located in remote environments since we have carefully chosen those without anthropogenic interferences and away from cultivated vineyards. As stated in the introduction, these vines are dioecious varieties with anemophilous pollination, thus not all years grape berries were present in the vines. Moreover, we pointed out in the first paragraph of the Results section that not all years was possible to collect grapes from the same vine plant due to drought, bird attacks, fungal infections of grapes or logistic concerns. Because of this, it was decided to group all the isolates into six regions: Azerbaijan, Georgia, North Italy, Romania, Sardinia (Italy) and Spain. Due to the great difference in samples collected from different regions, and additionally the different species richness in the analyzed regions, Hill’s number, specifically number of abundant species (N1), number of very abundant species (N2) and evenness (E’) were also calculated. Thus, grouping sampled places and applying these indexes we minimized the sampling biases. We have clarified this re-writing the first paragraph of the Introduction and adding a sentence in the Statistical analysis section as well as at the beginning of the Results section. We have added a new Supplementary Table to indicate the sites and the vine plants numbers sampled in each one.

  1. It is not clear how many vine plants were sampled at each site. The method description implies that only one sample (0.5-1.0 kg grape bunches) from one plant was sampled in each site. If it was true, the yeast diversity in each site was not fully revealed.

Thanks to the reviewer for pointing it out. In addition to the reasons remarked above, it was so difficult to access to the wild vine (they are lianas) and in the vast majority of the locations they are in rough terrains and occur in disjunct populations. The number of sampled vines was between 4-11 individuals in each location, depending on the grape presence or absence and the conditions. We have added this information at the beginning of the Introduction section and we have clarified the number of vine samples in a new Table as Supplementary material S1.

  1. The yeast strains isolated were differentiated mainly based on ITS RFLP analysis. This is an outdated method as mentioned in this manuscript that some species cannot be distinguished based on the RFLP patterns. Since DNA sequencing is commonplace and cheap now, why the authors did not use ITS and D1/D2 sequencing as the main method for yeast strain differentiation and species identification?

Yeasts identification was performed applying the ITS-PCR and the RFLP technique combined with the ITS and the LSU D1/D2 sequence analyses as the reviewer stated above. Actually, this method helps us to distinguish some misidentification in some of the species. At this moment this method is no outdated since it was included in the RESOLUTION OIV-OENO 408-2011 (International Organization of Vine and Wine) as molecular tool for identification of Saccharomyces cerevisiae wine yeast and other yeast species related to winemaking (please, see https://www.oiv.int/public/medias/1330/oiv-oeno-408-2011-en.pdf) and many researchers continue to use it. The main reasons we applied this method in combination with amplicon sequencings are because Aureobasidium pullulans can easily be confused with A. subglaciale, Kabatiella microsticta, Columnospaeria fagi because many database sequences of these species have identical D1/D2 sequences (e.g. PMID: 24984952, PMID: 26973603). Metschnikowia fructicola cannot be identified by rDNA analysis because it has a non-homogenized rDNA array (https://doi.org/10.1371/journal.pone.0067384). Similar problems were encountered at the taxonomic identification of M. pulcherrima-like strains (https://doi.org/10.1016/j.ijfoodmicro.2015.07.034).  

The D1/D2 sequences of the type strains of Cryptococcus carnescens and Cr. victoria differ only by 1.8% (less than the 2% set by the authors as taxonomic identity criterion). Rhodotorula nothofagi is difficult to distinguish from Curvibasidium pallidicorallinum because certain database sequences of these species have identical D1/D2 sequences.

To remedy the above problems, we compared the D1/D2 sequences and ITS sequences of the isolated strains with those of the type strains of the species (most can be found in the CBS database). We have deposited the results in the NBCI database (please, see accession numbers in the Supplementary material S1). Thus, we used the RFLP method to reinforce the correct identification because the database curators do not check the submissions for the correctness of taxonomic affiliation.

We have included a new paragraph in the results section to clarify the reviewer concern.

  1. Climate parameters were not measured and recorded; thus, it is unable to analyze the data in terms of ecology. It is not appropriate to call the data shown in Table 2 ecological indexes.

We agree with reviewer that climate parameters should be taken into consideration. But, as stated in the introduction and aims of the study, our main interest was the study of the diversity of culturable yeasts isolated from grape-berries from wild vines. Ecological indexes formulations applied in this study do not take into consideration the climatic parameters. Table 2 reports correctly the results obtained after the application of the ecological indexes on the basis of the total (meaning across all the sampled regions) frequency of isolation of the most abundant yeast species and their presence/absence in the different regions. Due to the large-scale sampling in different regions of the Mediterranean Basin, and the number of years, climate parameters was not possible to be recorded since some of the countries where samples were collected did not provide us the data from the meteorological stations. Moreover, wild vines grow in very special habitats where microclimatic conditions are present. We have added a sentence clarifying this issue in the introduction (First paragraph). We have deleted the sentence to minimize the climatic conditions from the first paragraph of the Results section to avoid confusion to the readers.

  1. It is questionable to include the yeast strains isolated after enrichment incubation in biodiversity index calculation and comparison as shown in Table 2, because enrichment must alter the yeast diversity originally harbored in the samples.

We agree that self-enrichment fermentations cannot be correctly extrapolated to evaluate the variations on berry microbiota. But the natural self-enrichment step is the only widely accepted method in the applied microbiology and enology areas since fermentative yeasts (mainly S. cerevisiae) take part of the must after a higher concentration of sugars (carbon source) and nitrogen, acidification of the media and a minimum content of ethanol in the must. These fermentative yeasts are naturally present in the must as viable but not culturable yeast at the beginning of the fermentation (please see; https://doi.org/10.1371/journal.pone.0077600 and https://doi.org/10.3389/fmicb.2016.00502). Because of this the auto-enrichment is strongly necessary to study the S. cerevisiae populations. We have clarified this concern in the M&M section as follows: For the detection of minority species which would not be detected by direct plating, an enrichment step repressing the propagation of alcohol-sensitive yeasts was also included in the process.

Furthermore, taking into account the latest taxonomic revision performed by Kurtzman et al. (2011) (https://theyeasts.org), who recognized an overall total of nearly 1500 yeast species belonging to 149 genera, only about 40 species were documented on grapes or in grape must in 2014 (Jolly,Varela and Pretorius 2014 https://doi.org/10.1111/1567-1364.12111) and around 150 in 2020 (Xu et al.2020 https://doi.org/10.1016/j.lwt.2019.108894). Thus, the self- enrichment strategy is strongly necessary to obtain all culturable yeast populations from them.

Moreover, we have added the new Supplementary Table separating yeasts strains from grapes and fresh and fermented musts, please see the new Supplementary Material.

  1. Isolation of culturable endophytic yeasts from the grape tissues is described in Materials and Methods, but the result is not mentioned in the Results section.

Thank you to point it out. We clarified in the M&M section that endophytic isolated yeasts were also considered as isolated from grape-berries fruits. Moreover, we have added additional information about this concern in both Results and Discussion sections as well as in the new Supplementary Table.

We thank the reviewer for help us to improve the manuscript. Substantial revisions have been made according to reviewer comments and recommendations.

Reviewer 2 Report

This work reported the description of  culturable yeast diversity of grapes from Vitis vinifera ssp sylvestris. A total of 3431 yeast colonies were collected from 2013 and 2016, in 5 countries and 29 sampling sites using different protocols to isolate epiphytic, endophytic and fermenting yeast populations. This study provides for the first time data on wine yeast assemblies associated with Vitis vinifera ssp sylvestris. The cultivation approach make it possible to build a yeast collection that could serve to investigate new biotechnological potentials.

However, before considering the paper for publication in Journal of Fungi,  I have some comments for the authors:

Introduction : last paragraph : the objective and the presentation of the study should be better explained.

Materials and methods

What is the maturity stage for grapes sampling? Then lane 184, page 4 : what was the protocol to evaluate the ripening status of the grapes?

Page 3, lanes 96-97 : the protocol described is efficient to collect epiphyte but not endophyte microorganisms.

Page 3, lanes 113-115 : do the authors used antibiotics against fungi or bacteria for yeast isolation?

Samples localization: table1, a map that summarized the samples localizations could be provided by the authors.

The number of colonies collected from grapes and enrichment is not clearly presented. Among 3431 colonies, how many were collected from grapes epiphytes, from grapes endophytes, from fermented must after 70 g/L of sugars consumption and at the end of alcoholic fermentation?

Do all must samples were positives for S. cerevisiae?

The use of only 3 microsatellite markers is not sufficient to established a robust genetic relationship between S. cerevisiae isolates. With higher number of loci, a dendrogram could have been built and clonality of the population analyzed.

Table 3 is not provided in the PDF file.

Discussion

Page 9,Lane 329: H uvarum is a dominant species associated with grape must but do not play an important role in grape juice fermentation in terms of population growth (due to its low ethanol tolerance). This part should be modulated since the positive role of H uvarum is not a general statement for wine microbiologists and enologists.

Author Response

This work reported the description of  culturable yeast diversity of grapes from Vitis vinifera ssp sylvestris. A total of 3431 yeast colonies were collected from 2013 and 2016, in 5 countries and 29 sampling sites using different protocols to isolate epiphytic, endophytic and fermenting yeast populations. This study provides for the first time data on wine yeast assemblies associated with Vitis vinifera ssp sylvestris. The cultivation approach make it possible to build a yeast collection that could serve to investigate new biotechnological potentials.

However, before considering the paper for publication in Journal of Fungi,  I have some comments for the authors:

Introduction : last paragraph : the objective and the presentation of the study should be better explained.

We thank to the reviewer for pointing it out. For a better understanding and presentation, we have added new information in the introduction section and we have re-written the last paragraph in the introduction as follows; “Under the best of our knowledge no studies focused on the study of yeast occurrence and diversity on carposphere of V. vinifera ssp. sylvestris from the Mediterranean Basin (29 different locations of Azerbaijan, Georgia, Italy, Romania and Spain) have been published before. Final outputs will allow: i) to obtain primary and precise information about yeast communities on berries of wild vines; ii) to provide an objective framework for the classification of the broadest range of species according to their extinction risk; iii) to select attractive yeast strains for their biotechnological potential, offering new opportunities to winemakers; iv) to set up a collection of wild strains with oenological origin in order to define novel domestication genetic targets for future evolutionary studies in wine yeasts.”

Materials and methods

What is the maturity stage for grapes sampling? Then lane 184, page 4 : what was the protocol to evaluate the ripening status of the grapes?

We thank the reviewer for highlighting this finding. The maturity stage of grapes during the sampling was performed first following the protocol proposed by Coombe (http://doi.wiley.com/10.1111/j.1755-0238.1995.tb00086.x) and then, on field, by simply applying a drop of 3 grape berries (randomly selected) onto the prism of a manual portable refractometer Master Series (ATAGO, Japan). Subsequently, the mean for each measurement was calculated. Measurements was expressed as °Brix. Afterwards, the samples characterized by a °Brix lower than 20 were considered as not-ripe, while the samples with a °Brix higher than 20 were considered as ripe. This methodology has been added in the M&M section.

Page 3, lanes 96-97 : the protocol described is efficient to collect epiphyte but not endophyte microorganisms.

We agree with the reviewer comment. We have clarified the protocol used to collect endophyte microorganisms in the M&M section. We have added the following reference: V. Isaeva, A.M. Glushakova, S.A. Garbuz, A.V. Kachalkin, I.Yu. Chernov, 2010, published in Izvestiya Akademii Nauk, Seriya Biologicheskaya, 2010, No. 1, pp. 34–43.

Page 3, lanes 113-115 : do the authors used antibiotics against fungi or bacteria for yeast isolation?

Thanks to the reviewer for pointing it out. We have detailed the composition of the YGC culture medium in the M&M section. We choose this medium since Chloramphenicol is an autoclavable antibiotic. Moreover, we have added the manufacturer name of the Wallerstein Laboratory commercial medium.

Samples localization: table1, a map that summarized the samples localizations could be provided by the authors.

We agree with reviewer suggestion. We have changed Table 1 as a map. We have moved coordinates and altitude of each location to Table S1.

The number of colonies collected from grapes and enrichment is not clearly presented. Among 3431 colonies, how many were collected from grapes epiphytes, from grapes endophytes, from fermented must after 70 g/L of sugars consumption and at the end of alcoholic fermentation?

Thank you to point it out. We clarified in the M&M section that endophytic isolated yeasts were also considered as isolated from grape-berries fruits. We have added additional information in both Results and Discussion sections about it. Moreover, we have provided a new Supplementary Material (now Table S1) according to the reviewer suggestions.

Do all must samples were positives for S. cerevisiae?

All musts were positive for non-Saccharomyces and S. cerevisiae after self-enrichment. We have added this information in the text (Results Section) and in the new Supplementary Table S1.

The use of only 3 microsatellite markers is not sufficient to established a robust genetic relationship between S. cerevisiae isolates. With higher number of loci, a dendrogram could have been built and clonality of the population analyzed.

The amplification of these three microsatellite loci was carried out according to the studies by González Techera et al. (2001), optimized by Vaudano and García Moruno (2008) and improved by Vaudano et al., 2019. In all these studies, it was used to identify and distinguish between different wine strains of S. cerevisiae. Since then, this technique has been used in different studies (González et al., 2022 (https://doi.org/10.1016/j.lwt.2022.113157); Ruiz-Muñoz et al., 2020; Vaudano et al., 2019; Marín-Menguiano et al., 2017; Francesca et al., 2014; Suzzi et al., 2012; Tello et al., 2012 or Cordero-Bueso et al., 2011). Undoubtedly, a different experimental approach would be necessary to decipher and to established a robust genetic relationship between S. cerevisiae isolates. But it is important to highlight that the aim in this work was to discern between different strains present in the different grapes and musts samples obtained from the different wild vine populations, and not to analyze the genetic relationship (which could have been done by analyzing and sequencing 10 or 12 loci). The technique used seems to be successful since 154 strains also showed phenotypic differences, and they have been detected throughout the samples during these four years of study. Thus, we considered this method as a good compromise between discriminatory ability, time required and cost-effective. Nevertheless, we have justified it in the Discussion section.

1Table 3 is not provided in the PDF file.

We are sorry for the mistake. Thank you for pointing it out. Maybe some technical issues between the word and pdf building and/or the uploading process of the documents have been occurred. We have included again the Table 3 in the document.

Discussion

Page 9,Lane 329: H uvarum is a dominant species associated with grape must but do not play an important role in grape juice fermentation in terms of population growth (due to its low ethanol tolerance). This part should be modulated since the positive role of H uvarum is not a general statement for wine microbiologists and enologist.

Thanks to the reviewer to pointing it out. We have modified the entire paragraph in the discussion section focusing the occurrence and diversity of all non-conventional yeast species found in the vine V. vinifera ssp. sylvestris with studies based on cultivated vines.

We thank the reviewer for help us to improve the manuscript. Substantial revisions have been made according to reviewer comments and recommendations.

Reviewer 3 Report

This paper reports on yeast collections from wild grapes in several regions of Europe.

The text is often ambiguous.  In particular, it can be difficult to evaluate accuracy and appropriate use of ecological statistics as the writing requires a major revision by someone with strong skills in English writing.

More discussion is needed on the significance of the yeast species recovered.  Are most of these species known to occur in association with grapes and related substrates?  “The Yeasts” should be consulted for natural history background on the major species.

To what extent can Vitis vinifera L. ssp. sylvestris be regarded as a ‘pristine environment’?

I note that a number of the results reported here have appeared previously
https://doi.org/10.1051/bioconf/20170902019, but the relevant reference is not given.

L.27,34,43 Caucasus (Caucasian is used for people, not geography).  Also the sentence needs to be re-written with proper punctuation, so as not to suggest that the Iberian peninsula extends all the way to India.

L.29 evidence of what?

L.85 An acknowledgement is made to a Dr. Maghradze.  Is that the same person who is listed as a co-author?  If so, the reference should be re-phrased.  Authors are generally not acknowledged within a paper.

L.158 – It is not clear what the authors mean by “the sequences of the type strains were also determined”.

L.193 “In total, 3431 culturable yeast colonies”  It would be more important to indicate how many plates were positive for yeast colonies and sampled from.  One could have isolated 3431 colonies from just a few plates, and that would greatly affect the significance of the results.  How many colonies were generally recovered from a plate?  How many replicates of each morphotype were purified, etc.?

L.221 “the lowest diversity was found in Azerbaijan”  For this to be meaningful, we should know how many samples were examined.  If one collects only one sample in a locality, the apparent species richness is by necessity going to be low.

Check spelling carefully: Isaatchenkia siamensis -  Aureubasidium proteae

In addition, I cannot see an occurrence of Issatchenkia siamensis in MycoBank.  The correct designation as given in GenBank should be used.  Also, it does not appear in the figure.

L.270 Spelling of Hanseiaspora

L.273 Spelling of Lanchaea thermotolerans and M. caribicca

L.343 It would be useful to know what is the closest relative of the yeast named Metschnikowia sp.

L.346 “S. cerevisiae had been isolated in musts and grape skins from all regions except Sardinia” to fully appreciate the significance of this finding, one would need to know whether the wild grapes occurred in the vicinity of active vineyards that might act as sources of domestic yeasts.  We are told very little about the general surroundings of the sampling areas.

L.361 What is meant by “The heterozygosity values did not coincide”?

L.363 “This phenomenon, which consists of a heterozygosity deficit, is related to genetic drift and non-random mating between individuals of the populations, related in turn to migration [56].”  this very general and vague, and I am not sure that the reference given is relevant.  Explain better or delete.

Author Response

This paper reports on yeast collections from wild grapes in several regions of Europe.

The text is often ambiguous.  In particular, it can be difficult to evaluate accuracy and appropriate use of ecological statistics as the writing requires a major revision by someone with strong skills in English writing.

We appreciate the reviewer’s comments. Figure 3 was missing in the previous version. We understand the concerns of the reviewer. Thus, we have newly included the Figure 3. Moreover, substantial revisions through all sections of the manuscript and English grammar have been made according to reviewer comments and recommendations. Please see the new version of the paper.

More discussion is needed on the significance of the yeast species recovered.  Are most of these species known to occur in association with grapes and related substrates?  “The Yeasts” should be consulted for natural history background on the major species.

We thank to the reviewer to help us to enhance the manuscript. We have significantly improved the paper adding more information in the Discussion section. Please, see the new version of the manuscript.

To what extent can Vitis vinifera L. ssp. sylvestris be regarded as a ‘pristine environment’?

We agree with reviewer’s comment. We have changed the word pristine by natural.

I note that a number of the results reported here have appeared previously
https://doi.org/10.1051/bioconf/20170902019, but the relevant reference is not given.

We agree with reviewer’s concern. We have shown before some preliminary results about the present work. These results were presented at the 40th World Congress of Vine and Wine as an oral conference. Then, preliminary results were published in the BIO Web of Conferences as a conference proceeding. We decided not to cite this document because as stated before, these are preliminary data, the number of isolates presented in the current manuscript is higher, so the identified species and the statistical analysis have significantly changed. Furthermore, the published conference proceeding is in Spanish language instead in the most used scientific language (English). Now we present a more robust and precise analysis of the data. We believe that the current document updates the conference proceeding. Nevertheless, we have added the Bio Web of Sciences report as reference in the introduction section (last paragraph).

L.27,34,43 Caucasus (Caucasian is used for people, not geography).  Also the sentence needs to be re-written with proper punctuation, so as not to suggest that the Iberian peninsula extends all the way to India.

We are sorry for the mistakes. We have corrected these errors through the manuscript.

L.29 evidence of what?

Thanks to the reviewer for pointing this out. We have corrected the sentence in the Introduction section as follows: Palynological evidences of pollen from the genus Vitis in the Middle Pleistocene were found in the ‘El Padul’ peat bogs in Granada, Spain…..

L.85 An acknowledgement is made to a Dr. Maghradze.  Is that the same person who is listed as a co-author?  If so, the reference should be re-phrased.  Authors are generally not acknowledged within a paper.

We agree with reviewer. We have deleted the names of the persons.

L.158 – It is not clear what the authors mean by “the sequences of the type strains were also determined”.

Thanks to the reviewer for the comment. We have altered the text in this paragraph of the Material and Methods section for a better explanation.

L.193 “In total, 3431 culturable yeast colonies”  It would be more important to indicate how many plates were positive for yeast colonies and sampled from.  One could have isolated 3431 colonies from just a few plates, and that would greatly affect the significance of the results.  How many colonies were generally recovered from a plate?  How many replicates of each morphotype were purified, etc.?

We really appreciate the comments provided by the reviewer that were very helpful for the improvement of our manuscript. We clarified in the M&M section that endophytic isolated yeasts were also considered as isolated from grape-berries fruits. We have added the number of colonies collected from grapes and enrichment. We have added additional information in both Results and Discussion sections about it. Moreover, we have added a new table as Supplementary Material to complete the reviewer suggestions (Table S1).

L.221 “the lowest diversity was found in Azerbaijan”  For this to be meaningful, we should know how many samples were examined.  If one collects only one sample in a locality, the apparent species richness is by necessity going to be low.

We agree with the reviewer’s comment, but samples were taken in natural environments where biotic and abiotic factors can’t be avoided. Number of samples collected from each location have been detailed in the M&M section as well as in the new Supplementary Table S1. Moreover, we pointed out in the first paragraph of the result section that not all years was possible to collect grapes from the same vine plant due to drought, bird attacks, fungal infections of grapes or logistic concerns. Because of this, it was decided to group all the isolates into six regions: Azerbaijan, Georgia, North Italy, Romania, Sardinia (Italy) and Spain. Due to the great difference in samples collected from different regions, and additionally the different species richness in the analyzed regions, Hill’s number, specifically number of abundant species (N1), number of very abundant species (N2) and evenness (E’) were also calculated. Thus, grouping sampled places and applying these indexes we minimized the sampling biases. We have clarified this re-writing the first paragraph of the Introduction and adding a sentence in the Statistical analysis section as well as at the beginning of the Results section.

Check spelling carefully: Isaatchenkia siamensis -  Aureubasidium proteae

We are sorry because of these mistakes. We have checked spelling of all species through the manuscript.

In addition, I cannot see an occurrence of Issatchenkia siamensis in MycoBank.  The correct designation as given in GenBank should be used.  Also, it does not appear in the figure.

We agree with reviewer that Issatchenkia siamensis do not appear now in Mycobank. Revising the designation in GenBank, it has changed since we deposited the sequence in 2020. Thus, we have decided to re-sequence twice the ITS and D1/D2 amplicons of this yeast strain. Now, these sequences could belong to Pichia occidentalis according to Mycobank and Genbank. We have deposited the new sequences in Genbank Accession numbers: ON076418 and ON076418. We have changed the names in the text and in the figures.

L.270 Spelling of Hanseiaspora

L.273 Spelling of Lanchaea thermotolerans and M. caribicca

We are sorry because of these mistakes. We have checked spelling of species through the manuscript.

L.343 It would be useful to know what is the closest relative of the yeast named Metschnikowia sp.

We agree with the reviewer comment. On the other hand, we have been problems with finding that some of the isolates which seem to belong to Metschnikowia-like strains, did not show sequence identity of their D1/D2 to any of the type strains despite they were fairly similar to one species of the Metschnikowia pulcherrima clade. It happened also with the ITS sequences. In agreement with species belonging to the so-called M. pulcherrima-clade cannot be unequivocally assigned to one of the species after rDNA analysis because some species such as M. fructicola or Metschnikowia andauensis have a non-homogenized rDNA array. When we compared the sequences of our isolates named as Metschnikowia sp. with the sequences deposited at the Mycobank database, the most probable species related to this genus were Metschnikowia chrysoperlae (similarity of 99.43%) and Metschnikowia pulcherrima (Similarity of 99.20%). Thus, we decided to maintain these sequences as Metschnikowia pulcherrima-like species. Same for the Hanseniaspora sp. species which matched as H. uvarum (lower than a 89% of similarity) in the Mycobank database. GenBank accessions numbers for all species are also provided in Table S1.

L.346 “S. cerevisiae had been isolated in musts and grape skins from all regions except Sardinia” to fully appreciate the significance of this finding, one would need to know whether the wild grapes occurred in the vicinity of active vineyards that might act as sources of domestic yeasts.  We are told very little about the general surroundings of the sampling areas.

Thanks to the reviewer for pointing this out. We have clarified in the M&M section that endophytic isolated yeasts were also considered as isolated from grape-berries fruits. We have added additional information in both Results and Discussion sections about it. Moreover, we have added three new columns to Table S1 (Supplementary Material) to complete the reviewer suggestions. Field investigations were carefully planned, choosing habitats with appropriate ecological characteristics for wild grapevine; well apart from today’s vineyards, proximity of rivers, soil with a high degree of humidity, and fairly good conservation level of the vegetation. We have included this information at the beginning of the M&M section.

L.361 What is meant by “The heterozygosity values did not coincide”?

L.363 “This phenomenon, which consists of a heterozygosity deficit, is related to genetic drift and non-random mating between individuals of the populations, related in turn to migration [56].”  this very general and vague, and I am not sure that the reference given is relevant.  Explain better or delete

We thank the reviewer for highlighting this finding. We have added Table 3 and modified the Discussion section for a better understanding and to solve the queries about the heterozygosity proposed by the reviewer. Please, see the new version.

We thank the reviewer for help us to improve the manuscript. Substantial revisions have been made according to reviewer comments and recommendations.

Round 2

Reviewer 1 Report

I thank the authors for their careful revision of the manuscript. The majority of my concerns have been accommodated in the revised version. I have only one minor comment on the revised version: the sampling locations in Figure 1 are not clear enough. 

Author Response

I thank the authors for their careful revision of the manuscript. The majority of my concerns have been accommodated in the revised version. I have only one minor comment on the revised version: the sampling locations in Figure 1 are not clear enough. 

We thanks to the reviewer for the encouraging words. We decided to add a map since another reviewer suggested it. We believe that a map is more impressive to readers. We have now added a number to each location from 1 to 29 and we have added the correspondence of each number to the name of each location in the figure caption. The full name, province, country and coordenates of each location can be found in the Supplementary Material.

Reviewer 3 Report

The authors have made a number of additions and corrections related to my suggestions.  Regarding the Metschnikowia species, it can be called M. pulcherrima and the recent paper by Sipicski can be cited -https://www.mdpi.com/2673-6500/2/1/9

My recommendation to have the complete text edited for English writing still needs to be accommodated.

Author Response

The authors have made a number of additions and corrections related to my suggestions.  Regarding the Metschnikowia species, it can be called M. pulcherrima and the recent paper by Sipicski can be cited -https://www.mdpi.com/2673-6500/2/1/9

We have clarified it in the result section and we have added a paragraph in the discussion section about this concern. We have also added the reference.

My recommendation to have the complete text edited for English writing still needs to be accommodated.

We have sent the paper to the English Editing Services of MDPI. We are now presenting the English-revised version of the manuscript. Please find also attached the English-Editing certificate
